# Full-Length Transcriptome Reconstruction Reveals the Genetic Mechanisms of Eyestalk Displacement and Its Potential Implications on the Interspecific Hybrid Crab (*Scylla serrata* ♀ × *S. paramamosain* ♂)

**DOI:** 10.3390/biology11071026

**Published:** 2022-07-07

**Authors:** Shaopan Ye, Xiaoyan Yu, Huiying Chen, Yin Zhang, Qingyang Wu, Huaqiang Tan, Jun Song, Hafiz Sohaib Ahmed Saqib, Ardavan Farhadi, Mhd Ikhwanuddin, Hongyu Ma

**Affiliations:** 1Guangdong Provincial Key Laboratory of Marine Biotechnology, Shantou University, Shantou 515063, China; spye@stu.edu.cn (S.Y.); 19xyyu@stu.edu.cn (X.Y.); 19hychen2@stu.edu.cn (H.C.); zhangyin@stu.edu.cn (Y.Z.); qywu1@stu.edu.cn (Q.W.); hqtan@stu.edu.cn (H.T.); 19jsong@stu.edu.cn (J.S.); saqib@stu.edu.cn (H.S.A.S.); farhadi@stu.edu.cn (A.F.); 2STU-UMT Joint Shellfish Research Laboratory, Shantou University, Shantou 515063, China; ikhwanuddin@umt.edu.my; 3Institute of Tropical Aquaculture and Fisheries, Universiti Malaysia Terengganu, Kuala Nerus, Terengganu 21030, Malaysia

**Keywords:** *Scylla* spp., interspecific hybridization, differential expression analysis, SMRT, illumina RNA sequencing

## Abstract

**Simple Summary:**

The eyestalk is a key organ in crustaceans that produces neurohormones and regulates a range of physiological functions. Eyestalk displacement was discovered in some first-generation (F1) offspring of the novel interspecific hybrid crab (*Scylla serrata* ♀ × *S.* *paramamosain* ♂). To uncover the genetic mechanism underlying eyestalk displacement and its potential implications, high-quality transcriptome was reconstructed using single-molecule real-time (SMRT) sequencing. A total of 37 significantly differential alternative splicing (DAS) events (17 up-regulated and 20 down-regulated) and 1475 significantly differential expressed transcripts (DETs) (492 up-regulated and 983 down-regulated) were detected in hybrid crabs with displaced eyestalks (DH). The most significant DAS events and DETs were annotated as being endoplasmic reticulum chaperone BiP and leucine-rich repeat protein lrrA-like isoform X2. In addition, the top ten significant gene ontology (GO) terms were related to the cuticle or chitin. Overall, this study highlights the underlying genetic mechanisms of eyestalk displacement and provide useful knowledge for mud crab (*Scylla* spp.) crossbreeding.

**Abstract:**

The lack of high-quality juvenile crabs is the greatest impediment to the growth of the mud crab (*Scylla paramamosain*) industry. To obtain high-quality hybrid offspring, a novel hybrid mud crab (*S. serrata* ♀ × *S. paramamosain* ♂) was successfully produced in our previous study. Meanwhile, an interesting phenomenon was discovered, that some first-generation (F1) hybrid offspring’s eyestalks were displaced during the crablet stage I. To uncover the genetic mechanism underlying eyestalk displacement and its potential implications, both single-molecule real-time (SMRT) and Illumina RNA sequencing were implemented. Using a two-step collapsing strategy, three high-quality reconstructed transcriptomes were obtained from purebred mud crabs (*S. paramamosain*) with normal eyestalks (SPA), hybrid crabs with normal eyestalks (NH), and hybrid crabs with displaced eyestalks (DH). In total, 37 significantly differential alternative splicing (DAS) events (17 up-regulated and 20 down-regulated) and 1475 significantly differential expressed transcripts (DETs) (492 up-regulated and 983 down-regulated) were detected in DH. The most significant DAS events and DETs were annotated as being endoplasmic reticulum chaperone BiP and leucine-rich repeat protein lrrA-like isoform X2. In addition, the top ten significant GO terms were related to the cuticle or chitin. Overall, high-quality reconstructed transcriptomes were obtained for the novel interspecific hybrid crab and provided valuable insights into the genetic mechanisms of eyestalk displacement in mud crab (*Scylla* spp.) crossbreeding.

## 1. Introduction

The mud crab (*Scylla* spp.) is a commercially important aquaculture species in Southeast Asian countries, mainly distributed along the coasts of India and the Western Pacific. In the last decade, to meet the high-quantity protein demand of humans, the production of mud crab farming has increased rapidly, but the majority of mud crab farming still relies on wild-caught seed crabs [1]. However, both the quantity and quality of mud crab seeds in the wild have decreased dramatically due to over-exploitation and environmental deterioration. Currently, the lack of high-quality juvenile crabs is considered as the main challenge in the expansion of the mud crab (*S. paramamosain*) industry. Furthermore, the genetic improvement of mud crabs (*S. paramamosain)* is still in its infancy compared with other aquatic species. Most studies focus on intraspecies classification [2,3], sex determination [4,5,6], sex identification [7,8], and nutritional composition [9,10]. Therefore, it is critical to speed up the genetic improvement of the mud crab (*S. paramamosain*).

Interspecific hybridization is a common and effective method for genetic improvement and breeding of aquatic animals [11] because hybrid offspring usually have more biomass, faster growth, and higher fertility than both parents [12]. To obtain a high-quality hybrid offspring, a novel hybrid mud crab (*S. serrata* ♀ × *S. paramamosain* ♂) was successfully developed in our previous study [13,14]. However, interspecific hybridization in the mud crab (*Scylla* spp.) not only brings hybrid vigor but also hybrid inferiority, because some hybrid offspring had eyestalk displacement during the crablet stage I. Compared with normal eyestalks, displaced eyestalks were located in the center of the head and extended forward close to the antenna (detailed morphological features and photographs were shown in our previous study [15]). As we know, the eyestalk is an important neuroendocrine organ complex and controls a variety of physiological processes involving the central pacemakers of circadian rhythms [16], osmotic regulation [17], molting [18], and reproduction in crustaceans [19]. Most importantly, molting and reproduction are the essential processes to improve productivity in crustacean aquaculture. As the awareness of crustacean endocrinology has increased, eyestalk ablation has become an effective method for promoting molting and reproduction in crustacean based on the removal of the gonad- and molt-inhibiting hormones [20,21]. Eyestalks plays an important role in crustacean aquaculture. Therefore, it is essential to investigate and understand the genetic mechanisms behind eyestalk displacement and its potential implications on the novel hybrid mud crab. 

Over the past decade, with the rapid development of high-throughput sequencing technology, transcriptome analysis based on next-generation sequencing (NGS) has been widely implemented to identify the genetic mechanisms underlying the complex biological processes in crustacean studies [22]. However, the read lengths of NGS approaches are relatively short and limit the genome assembly in complex regions, especially for species without a reference genome (such as hybrids) [23]. To address this issue, the single-molecule real-time (SMRT) sequencing technology has been developed to improve the read length of sequencing with high accuracy [24]. Moreover, the sequencing process of SMRT occurs in real-time and doesn’t need PCR amplification during sequencing and library preparation, reducing the PCR related bias. Nowadays, full-length transcriptome sequencing has been successfully implemented in crustaceans to investigate the gene expression patterns of specific tissues or life stages and conditions in a transcriptomic experiment, such as *Litopenaeus vannamei* [25] and *S. paramamosain* [26]. Therefore, using full-length transcriptome sequencing data to reconstruct the transcriptomes of the hybrid crabs would help uncover the underlying genetic mechanisms of eyestalk displacement.

In this study, the transcriptomes of hybrid crabs with displaced eyestalks (DH), hybrid crabs with normal eyestalks (NH), and purebred mud crabs (*S. paramamosain*) with normal eyestalks (SPA) were reconstructed using PacBio sequencing data. Following that, novel genes, transcript isoforms, and alternative splicing (AS) were identified and annotated. Furthermore, the differential expression transcripts (DETs), differential alternative splicing (DAS) events, and their enrichment analysis between DH and NH were performed using Illumina RNA sequencing data. Finally, several DETs and DASs were selected for further validation of RNA-seq data. These results will facilitate highlighting the underlying genetic mechanisms of eyestalk displacement and provide useful knowledge for mud crab (*Scylla* spp.) crossbreeding.

## 2. Material and Methods

### 2.1. Sample Collection, RNA Extraction and Sequencing

The F1 hybrid offspring were developed in our previous study using *S. paramamosain* as the male parent and *S. serrata* as the female parent [13,14]. In brief, the male *S. paramamosain* was obtained from the local shore (Guangdong province, China), and the female *S. serrata* was bought from a local market in Shantou, China. The artificial mating of *Scylla* was performed following the method described by our previous study [13]. After artificial mating, female crabs were transferred to a rearing tank until the egg hatching. The F1 hybrid larvae were hatched in circular fiberglass tanks (0.9 m diameter, 1.0 m height), and then transferred to concrete rearing tanks (5.8 m × 4.8 m × 1.8 m) for growing seedlings. Culture conditions were ambient temperature (almost 30 °C), a natural photoperiod, and salinity of approximately 30 ppt. Samples of DH, NH, and SPA were collected for SMRT and Illumina sequencing at the first stage of the crablet. Briefly, these juvenile crabs were anaesthetized in ice cold water for 5 min before being snap-frozen in liquid nitrogen and stored at −80 °C for RNA extraction. The total RNA was then extracted from the whole-body using RNA isoPlus (TaKaRa, Shiga, Japan) following the manufacture’s instruction. Furthermore, the RNA quality was assessed in terms of integrity, purity, and concentration using Agilent 2100 Bioanalyzer (Agilent Technologies, CA, USA) and Nanodrop 2000 (Thermo Fiser Scientific, CA, USA). Finally, only high quality RNA was used to construct libraries and sequencing. The library preparation and sequencing were carried out at the Beijing Novogene Bioinformatics Technology Co. Ltd., China. For SMRT sequencing, three libraries for DH, NH, and SPA were constructed and sequenced on the PacBio Sequel platform, respectively. In addition, each SMRT library included pooled RNA from three crab samples. For Illumina sequencing, three libraries of the DH group (one abnormal hybrid crab per library) and three libraries of the NH group (two normal hybrid crabs per library) were constructed and sequenced on the Illumi-na Hiseq 2500 platform to generate 150 bp paired-end reads.

### 2.2. SMRT Sequencing Data Processing

To obtain transcript isoforms, the subread sequences data were processed with the following five steps: (1) The subread sequence data were processed by the ccs v4.2.0 software with the parameters —minLength 50, —maxLength 15,000, —minPasses 1, to generate circular consensus sequences (CCS); (2) the full-length (FL) reads were obtained from CCS by primer removal and demultiplexing using the lima v1.11.0 software with the parameters —dump-clips and —peek-guess; (3) the noise of FL reads was removed using the refine module of the isoseq3 v3.3.0 software with the parameters —require-polya and —min-polya-length 20; and (4) the consensus sequences from the same transcript were clustered to generate unpolished transcripts using the refine module of the isoseq3 v3.3.0 software with default parameters; and finally (5), the unpolished transcripts were polished to yield high-quality and low-quality isoforms using the polish module of the isoseq3 v3.3.0 software with default parameters.

### 2.3. Collapsing Redundant Transcripts Isoforms

To eliminate the redundant transcript isoforms, a two-step collapsing strategy was used in this study. In short, the high-quality isoforms were firstly aligned and sorted to generate SAM files by minimap2 (version 2.18) [27] with default parameters using the mud crab (*S. paramamosain*) as the reference genome [28]. Based on the mapping results, redundant isoforms were collapsed by the cDNA cupcake software (https://github.com/Magdoll/cDNA_Cupcake, accessed on 21 June 2021) with the parameters —min_aln_coverage 0.95, —min_aln_identity 0.85, and —dun-merge-5-shorter. In addition, unmapped transcripts were also collapsed using the Cogent v8.0 software (https://github.com/Magdoll/Cogent, accessed on 21 June 2021) and cDNA cupcake software with default parameters. In this process, different gene families were discovered initially from these unmapped transcripts. Then, a “fake genome” was created by concatenating all cogent unassigned contigs. Using the “fake genome” as the reference genome, these unmapped transcripts were collapsed by the cDNA cupcake software according to the abovementioned steps. Finally, CD-HIT (version 4.8.1) [29] was used to eliminate highly identical sequences from both mapped and unmapped transcript isoforms for further analysis. 

### 2.4. Completeness and Characteristics Analysis of Reconstructed Transcriptomes 

To evaluate the quality and completeness of the full-length transcriptomes, benchmarking universal single-copy orthologs (BUSCO) analysis were performed using BUSCO v5.1.3 software [30] with transcriptome mode and Arthropoda OrthoDB (arthropoda_odb10) [31] for full-length transcripts from DH, NH, and SPA. After determining the completeness, full-length transcripts were classified by comparing them to the reference genome annotation using gffcompare v0.12.2 software [32]. In this step, full-length transcripts were classified into 15 classes, including annotated (coded as “=” or “c”), novel isoform (coded as “j” or “k”), retrained intron (coded as “m” or “n”), novel antisense (coded as “x”), or novel intronic/intergenic (coded as “i” or “u”). The detailed information can be found on the official website of the gffcompare software (http://ccb.jhu.edu/software/stringtie/gffcompare.shtml, accessed on 6 February 2021). In addition, the transcriptome data of DH, NH, and SPA were paired for comparison to identify the common and unique transcripts using blastn v2.11.0+ and the following parameters: −*e* value 1 × 10^-10^ and −perc_identity 0.95. In this process, the common and unique transcripts between transcriptomes were identified by turning one transcriptome as a blast database and the other transcriptome as a query sequence.

### 2.5. Gene Functional Annotation

To have a better understanding of the biological context of the full-length transcripts, gene functional annotation was conducted. In summary, a TransDecoder was used to extract open reading frames (ORFs) from full-length transcripts using default parameters. If multiple ORFs were found in a single transcript, the first appeared ORF was selected for further analysis. The resulting ORFs were identified using eggNOG-mapper (v 2.1.4) [33] using the default parameters for obtaining this functional annotation information, such as the clusters of orthologous groups (COG), gene ontology (GO), Kyoto encyclopedia of genes and genomes (KEGG), and protein families database (Pfam). Additionally, the resulting ORFs were also searched against four databases (Swiss-Prot, TrEMBL, Uniref90, and NR) using the blastp function of diamond (v2.0.4.142) with the following parameters: -outfmt 6, -max_target_seqs 1, and −*e* value 1 × 10^−5^. Finally, all ORF annotation results were integrated and reported as a tab-delimited summary file.

### 2.6. Alternative Splicing (AS) Events Analysis

The alternative splicing (AS) events were evaluated using SUPPA2 software [34] utilizing a GTF annotation file obtained from DH, NH, and SPA to investigate the differences in gene expression patterns in purebred and crossbreed mud crabs. Using the generateEvents function of the SUPPA2 software with default parameters, seven AS event types were generated from the GTF annotation file, including SE (skipped exon), MX (mutually exclusive exon), A5 (alternative 5′ splice site), A3 (alternative 3′ splice site), RI (retained intron), AF (alternative first exon), and AL (alternative last exon). The distribution of AS events and the common overlaps were identified and visualized to compare the differences among DH, NH, and SPA. In addition, the GO enrichment analysis of AS events was performed by clusterProfiler using the results of eggNOG-mapper annotation.

### 2.7. Quantification of Identified Transcripts

For analyzing the differential alternative splicing events between NH and DH, the subreads sequences of DH and NH were merged, and processed to generate the reconstruction transcriptomes for the interspecific hybrid crab (*S. serrata* ♀ × *S. paramamosain* ♂) following the abovementioned steps. Using the reconstructed transcriptomes, the quantification of the expression of different transcripts and genes was estimated using Salmon with the mapping-based mode [35]. In this step, the mapping-based index of reference transcript sequences was initially constructed using an auxiliary k-mer hash over k-mers of length 31. Then, all clean data from RNA-seq were aligned to the reconstructed transcriptomes quicky and accurately using the quant module of Salmon with parameters—l IU and—validateMappings. Finally, the quantmerge module of Salmon was used to obtain transcripts per million reads (TPM) of each sample, which was then used to calculate additional inclusion levels (PSIs).

### 2.8. Differential Alternative Splicing (DAS) Events, Differential Expressed Transcripts (DETs), and Their Enrichment Analysis

To discover the differential alternative splicing (DAS) events between NH and DH, the inclusion levels (PSIs) per AS events were determined by the psiPerEvent function of SUPPA2 software using the results of TPM and AS events. Furthermore, significant differential expression analysis between NH and DH was performed by the diffSplice function of SUPPA2 software using this criterion |∆PSI| ≥ 0.15 and *p* value < 0.05. In addition, differential expression analysis between NH and DH was performed using DESeq2 [36] with their transcripts’ expression levels (TPM values). Significant differentially expressed transcripts (DETs) were identified using the threshold |log2(Fold Change)| ≥ 1 and *p* value < 0.05. Finally, for determining the functions of significant DAS events and DETs, GO and KEGG pathway enrichment analysis were performed. During the GO and KEGG enrichment analysis, the annotations of related genes were initially extracted from the results using eggNOG-mapper. Later on, the GO and KEGG enrichment analysis of these genes were performed by clusterProfiler [37]. Finally, only the GO terms or pathways which had *p* < 0.05 were denoted as significant.

### 2.9. Validation of AS Events and Differential Expressed Genes

To validate the transcriptome sequencing results, three genes with AS events were selected for RT-PCR and four genes from differential expressed gene analysis were selected for qRT-PCR. To summarize, total RNAs were used to synthesize the cDNA firstly using the GoScript™ Reverse Transcription System (Madison, WI, USA, Promega). Primer pairs of these nine genes were designed using Primer Premier 6.0 Software (Appendix A). For validation of AS events, the reverse transcription products of three genes were subjected to PCR analysis to obtain PCR products for agarose gel electrophoresis. To validate the differential expressed genes between NH and DH, the qRT-PCR of four genes were performed according to the manufacturer’s protocol in a LightCycler^®^ 480 system (Indianapolis, IN, USA, Roche Applied Science) using a miRcute Plus miRNA qPCR Kit (SYBR Green) (TIANGEN Biotech, Beijing, China) and Talent qPCR Premix (SYBR Green) kit (TIANGEN Biotech, Beijing, China). In this process, 18S rRNA was used as the internal control (reference genes), each gene was amplified in three biological replicates and three technical replicates, relative fold-change was calculated using the 2^−^^∆∆CT^ method [38], and a student’s *t*-test was used to determine the statistical significance (*p* < 0.05) using R software.

## 3. Results

### 3.1. Summary of PacBio Iso-Seq Data

We used PacBio SMRT sequencing on RNA samples extracted from NH, DH, and SPA at the crablet stage I to examine the expression patterns of interspecific hybrid crabs and its potential contribution to eyestalk displacement. A total of 12,705,473 (20.8 GB), 15,425,110 (23.0 GB), and 15,459,279 (22.22 GB) subreads for SPA, NH, and DH were recovered. These subreads yielded 265,745, 420,743, and 367,638 circular consensus sequences (CCS) for SPA, NH, and DH, respectively. Following the IsoSeq3 refinement, clustering, and polishing steps, a total of 11,643 (14), 16,587 (12), and 10,336 (9) high-quality (low-quality) isoforms with average lengths of 1780.5 (1883.8), 1630.5 (857.1), and 1661.6 (604.1) bp were obtained for SPA, NH, and DH, respectively. The proportion of low-quality isoforms in comparison to high quality isoforms was minimal so we could exclude them from further analysis (Table 1).

### 3.2. Collapsing Redundant Isoforms

After SMRT sequencing data processing, the high-quality isoforms still included a considerable number of redundant isoforms. In this study, a two-step collapsing strategy was applied to collapse redundant isoforms. Based on the results of the reference genome mapping, redundant isoforms were collapsed, generating 9427, 11,639, and 7858 unique isoforms in SPA, NH, and DH, respectively (Table 2). The remaining unmapped sequences were utilized to construct the “fake genome” to collapse redundant isoforms, yielding 466, 752, and 611 unique isoforms in SPA, NH, and DH, respectively (Table 2). Only a fraction of transcripts remained unmapped after collapsing redundant isoforms with a two-step strategy (Table 2). Finally, all unique isoforms were merged using CD-HIT to further collapse redundant isoforms, generating 9872, 12,382, and 8508 isoforms with average lengths of 1798.9, 1663.6, and 1685.5 bp for SPA, NH, and DH, respectively (Table 2). In general, a two-step collapsing strategy generates more unique isoforms in comparison with collapsing redundant isoforms based only on the reference genome.

### 3.3. Evaluation of Reconstructed Transcriptomes

The completeness and quality of transcriptome for SPA, NH, and DH are important prerequisites for further analysis. In this study, both completeness and characteristics analysis of reconstructed transcriptomes for SPA, NH, and DH were performed, and the results are shown in Figure 1. BUSCO assessment results showed that the number of complete and single-copy transcripts were 328 (32.4%), 302 (30.0%), and 315 (31.1%), duplicated transcripts were 161 (15.9%), 124 (12.2%), and 76 (7.5%), fragmented transcripts were 14 (1.4%), 19 (1.9%), and 16 (1.6%), and missing transcripts were 510 (50.3%), 568 (56.1%), and 606 (59.8%) for SPA, NH, and DH, respectively (Figure 1A). When overlapping transcripts across these three reconstructed transcriptomes were examined, the number of overlapping transcript isoforms for DH compared with SPA was 6983 and 7087 when DH was compared with NH, and 8823 for SPA compared with NH (Figure 1B). The number of unique transcript isoforms was 1525 or 1421 when DH was compared to NH or SPA, 5399 or 3559 when NH was compared to DH or SPA, and 2785 or 1421 when SPA was compared to DH or NH (Figure 1B). In a word, the number and diversity of transcript isoforms in NH samples were greater than in other samples, and that most transcript isoforms of DH (82%) and SPA (89%) were similar to NH. Moreover, these reconstructed transcriptomes had numerous isoforms, with the number of transcripts with more than two isoforms being 1756, 2050, and 1393 in SPA, NH, and DH, respectively (Figure 1C). In comparison to the reference genome annotation, a total of 1992, 2781, and 2566 potentially novel isoforms (coded as j) and a total of 3032, 4171, and 3297 unknown isoforms (coded as u) were annotated in reconstructed transcriptomes of DH, NH, and SPA, showing that the reconstructed transcriptomes contain more novel isoforms or transcripts. In addition, the number of transcripts isoforms that were annotated as other same strand overlaps with reference exons (coded as o) was 469 (3.8%) in NH, which was higher than DH 235 (2.8%) and SPA 240 (2.5%).

### 3.4. Functional Annotation

To obtain the biological context of the reconstructed transcripts, functional annotation was carried out for SPA, NH, and DH, respectively. Before functional annotation, a total of 8207, 9126, and 6323 ORFs were retrieved and selected from full-length transcripts in SPA, NH, and DH, respectively. That is, approximately 83.1%, 73.7%, and 74.3% of the transcript isoforms were potential protein-encoding segments of SPA, NH, and DH, respectively. The functional annotation results showed that the number of annotated transcript isoforms acquired from various databases varied, ranging from 5334 to 7856 in SPA (Figure 2A), 5273 to 8177 in NH (Figure 2B), and 3980 to 6023 in DH (Figure 2C). The number of transcript isoforms annotated at least once by Swiss-Prot, TrEMBL, Uniref90 or NR were 7871, 8210, and 6040 in SPA, NH, and DH, respectively (Figure 2D–F). The number of overlapping annotated transcript isoforms among Swiss-Prot, TrEMBL, Uniref90, and NR were 6163, 5823, and 4562 in SPA, NH, and DH, respectively (Figure 2D–F). In addition, transcript isoforms were classified into several COG categories to acquire a better understanding of gene functions enriched in SPA, NH, and DH (Figure 2G). The proportion of signal transduction mechanisms (COG category T) and cytoskeleton (COG category Z) in DH was comparable to that of NH but greater than SPA. NH also had a smaller proportion of carbohydrate transport and metabolism (COG category G) than DH and SPA. However, a contrasting pattern was observed in extracellular structures (COG category W) (Appendix A).

### 3.5. Alternative Splicing (AS) Events

AS events plays critical roles in the transcriptome diversity and complexity of eukaryotes. To explore the different AS events in purebred and crossbred mud crabs, AS analyses were performed in SPA, NH, and DH (Figure 3). The results revealed that the trend of the proportion of AS event types was similar between NH and SPA, with the most abundant RI (retained intron), followed by A3 (alternative 3′ splice site), A5 (alternative 5′ splice site), AF (alternative first exon), SE (skipped exon), MX (mutually exclusive exon), and AL (alternative last exon) (Figure 3A). The proportion of A5 in DH was lower than SE and AF (Figure 3A). Additionally, the total number of AS events in DH (1224) fell well below NH (1784) and SPA (1802). A total of 215, 232, and 233 overlapped AS events were found for DH compared with NH, DH compared with SPA, and SPA compared with NH, respectively, and 132 of these were common AS events among SPA, NH, and DH (Figure 3B). Moreover, there were 909, 1468, and 1486 unique AS events detected in SPA, NH, and DH, respectively (Figure 3B, Appendix A). To gain an insight into the function of these AS events, gene ontology (GO) enrichment analyses of all genes with AS events were performed in SPA, NH, and DH. In SPA, six of the top seven significantly biological processes (BP) obtained from the GO enrichment analysis were related to metabolic processes such as the NAD metabolic process, NADH metabolic process, pyridine-containing compound metabolic process, regulation of the viral process, pyridine nucleotide metabolic process, and nicotinamide nucleotide metabolic process (Figure 3C). However, for NH or DH, the top ten significant BP categories mainly occurred in muscle related to cell differentiation, muscle contraction, and the muscle system process (Figure 3D,E). In addition, when the top ten significant BP categories in NH and DH were examined, we found that chitin-based cuticle development and cuticle development were only significantly enriched in NH, suggesting that it may potentially contribute to abnormal eyestalks (Figure 3D,E).

### 3.6. Differential Alternative Splicing (DAS) Events, Differential Expressed Transcripts (DETs), and Their Enrichment Analysis

To identify and understand the potential implications of eyestalk displacement on the novel hybrid mud crab, both DAS events and DETs analysis between NH and DH were performed in this study. In the DAS event analysis, a total of 37 significant DAS events were found between NH and DH, with 17 up-regulated and 20 down-regulated in DH (Figure 4A). All these significant AS events were expressed by 28 protein-coding genes and included 8 A3, 7 A5, 1 AF, and 20 RI events (Appendix A). The top three significantly DAS genes experienced RI events, including the PB.2110 (endoplasmic reticulum chaperone BiP), PC.398 (troponin T), and PB.5116 (signal peptidase complex subunit 1-like). More detailed information is provided in Appendix A. In DET analysis, a total of 1475 significant DETs were found between DH and DH, comprising 492 up-regulated and 983 down-regulated transcript isoforms in DH (Figure 4B). The top ten significant transcripts based on *p*-values were PB.4294.2 (leucine-rich repeat protein lrrA-like isoform X2), PB.2110.3 (endoplasmic reticulum chaperone BiP), PB.821.4 (mite allergen Der f 3), PB.1808.5 (xylose isomerase-like isoform X2), PB.1547.11 (myosin heavy chain, muscle), PB.4122.4 (lysosomal Pro-X carboxypeptidase), PB.68.20 (cryptocyanin 1, partial), PB.5911.24 (hypothetical protein FQN60_009376), PB.2165.1 (mitochondrial-processing peptidase subunit alpha), and PB.1741.15 (aconitate hydratase, mitochondrial-like). However, the top ten significant genes based on fold change were PB.768.16, PB.1546.30 (myosin heavy chain, muscle), PB.3351.46 (tropomyosin slow-tonic isoform), PB.5762.10 (epidermal growth factor receptor kinase substrate 8), PB.3351.61 (tropomyosin slow-tonic isoform), PB.5203.12 (hypothetical protein GWK47_043851), PB.4579.4 (pro-resilin), PC.514.1 (glutathione S-transferase D7), PB.4574.4 (pro-resilin), and PB.753.1. Among, PB.5762.10 (epidermal growth factor receptor kinase substrate 8), PB.4579.4 (pro-resilin), and PB.4574.4 (pro-resilin) were closely linked to the epidermal development. More detailed information can be found in Appendix A. Furthermore, between significant DAS events and DETs, there were 21 overlapping and seven non-overlapping genes (Appendix A). To gain further insight into their underlying molecular mechanisms, GO and Kyoto Encyclopedia of Genes and Genomes (KEGG) enrichment analyses were performed using both DAS events and DETs. We found that majority of the top ten significantly enriched GO terms were related to the cuticle or chitin, including cuticle development, chitin-based cuticle development, structural constituent of cuticle, structural constituent of chitin-based larval cuticle, structural constituent of chitin-based cuticle, and chitin binding (Figure 4C and Appendix A). Furthermore, we found that these DAS events and DETs were primarily involved in RNA polymerase, protein digestion and absorption, exosome [BR:ko04147] peptidases and inhibitors [BR:ko01002], DNA repair and recombination proteins [BR:ko03400], and transcription machinery [BR:ko03021] (Figure 4D and Appendix A). When compared to the other pathways, the exosome pathway had the most genes (80) with significant levels (adjusted *p*-value = 0.0001).

### 3.7. Validation of Alternative Splicing (AS) Events and Significant Differential Expression Transcripts (DGEs)

To validate the predicted AS events by SMRT sequencing, two genes with SE events (PB.3871 and PB.5095) and one gene with RI events (PB.293) were selected randomly to perform reverse transcription PCR (RT-PCR). The primers were designed in overlapping exons for genes with SE events and retained intron for gene with RI event. The results of agarose gel electrophoresis showed that these AS events actually existed in DH, NH, and SPA, and the PB.5095 had a special transcript isoform in NH (Figure 5A). For validating the Illumina sequencing results, four genes (PB.3297, PB.3654, PB.2760 and PB.480) from differential transcript expression analysis between NH and DH were selected for quantitative real-time PCR (qRT-PCR) (Figure 5B). The results showed that PB.3297 had a similar expression level in DH and NH (*p* = 0.84), the expression level of PB.3654 in DH was significantly higher than in NH (*p* = 0.0001), and the expression level PB.2760 and PB.480 in DH had a significantly lower level than NH (*p* = 0.01 and 0.02) (Figure 5B and Appendix A). In general, the pattern of differentially expressed expression levels from the qRT-PCR results was consistent with the Illumina sequencing results. 

## 4. Discussions

Interspecific hybridization, which is well known for developing hybrid offspring with greater biomass, speed of development, and fertility, is considered one of the effective breeding technologies in aquatic animals [11]. In our earlier work, we successfully developed a novel hybrid mud crab (*S. serrata* ♀ × *S. paramamosain* ♂) for obtaining high-quality hybrid offspring for genetic improvement [13,14]. Meanwhile, we found that some F1 hybrid offspring’s eyestalks had displaced during the crablet stage I. However, the genetic mechanisms of eyestalk displacement and its potential impact on the physiological development of the novel interspecific hybrid crab (*S. serrata* ♀ × *S. paramamosain* ♂) remains unclear. In this study, we constructed PacBio and Illumina HiSeq libraries to reconstruct high-quality transcriptomes to detect and analyze the novel genes, transcripts isoforms, and AS events among SPA, NH, and DH. Moreover, DAS analysis and DET analysis between NH and DH, and their enrichment analyse were performed to examine the genetic mechanisms of eyestalk displacement and its potential implications on the novel interspecific hybrid crab.

### 4.1. Reconstructed Transcriptomes Based on the Non-Hybrid Correction Methods and Two-Step Collapsing Strategy

The use of SMRT sequencing data to reconstruct transcriptomes for improving genome annotation has been widely used in several species [26,39,40]. In comparison to NGS technology, SMRT sequencing could precisely capture each full-length transcript to eliminate assembly errors and identify novel transcript isoforms and AS events [41,42]. Although SMRT sequencing proved effective in capturing transcript structure, it has the following shortcomings: a high sequencing error rate (approximately 15%) and low sequencing throughput [43,44]. Previously, the hybrid correction method (combining the strengths of SMRT and Illumina RNA sequencing) was always used to correct the high SMRT sequencing error rate [45]. Nowadays, numerous non-hybrid correction methods that exclusively use long reads have been proposed for SMRT sequencing data processing [46]. In this study, the non-hybrid correction method was used to correct SMRT sequencing errors, and the results showed that only 14, 12, and 9 low-quality isoforms remained in SPA, NH, and DH, respectively (Table 1), indicating that the non-hybrid correction method also was suitable for SMRT sequencing error correction. Even after polishing, high-quality isoforms retained a large number of redundant transcript isoforms. In this study, in order to obtain high-quality transcriptomes of SPA, NH, and DH, a two-step collapsing strategy was deployed to collapse redundant high-quality isoforms. In comparison to the collapsed isoforms with the reference genome only, this strategy might boost the coverage and diversity of reconstructed transcriptomes (Table 2). Moreover, a total of 9872, 12,382, and 8508 isoforms with an N50 of 2079, 2041, and 2161 bp were obtained for SPA, NH, and DH, respectively (Table 2), which was longer than previous studies on *S. paramamosain* that reconstructed transcriptomes using NGS technologies [47,48]. However, the mean length and N50 of transcript isoforms from this study were much shorter than previous SMRT sequencing studies in mud crabs (*S. paramamosain*) [26,49,50]. The potential reason may be the use of a two-step strategy for collapsing redundant transcript isoforms, resulting in the retention of many short transcript isoforms (Table 2). Samples with different periods and tissues among these studies may be the other reason, because gene expression usually has tissue specificity and spatiotemporal specificity [51,52]. In summary, both the non-hybrid correction method and the two-step collapsing strategy were suitable for SMRT sequencing data processing and would provide more distinct isoforms.

### 4.2. The Genetic Mechanisms of Eyestalk Displacement and Its Potential Implications on the Novel Interspecific Hybrid Crab

The eyestalk is a key organ in crustaceans that produces neurohormones and regulates a range of physiological functions. In this study, both DAS events and DET analysis were deployed to explore the genetic mechanisms of eyestalk displacement and its potential implications on the novel interspecific hybrid crab. The most substantially annotated DET was leucine-rich repeat protein lrrA-like isoform X2. To the best of our knowledge, the leucine-rich repeats are protein interaction motifs with 20 to 29 residues that include a high proportion of leucine residues [53]. Most of the studies on leucine-rich repeat, motif-containing proteins were focused on the immunological response, in crustaceans, such as *S. serrata* [54], *Penaeus monodon* [55]*,* and *Litopenaeus vannamei* [56]. In other species, such as Drosophila, leucine-rich repeat, motif-containing proteins are also involved in cytoskeleton remodeling [57], cell morphogenesis [58], and segment morphogenesis [59]. We conclude that the leucine-rich repeat protein lrrA-like isoform X2 may play a critical role in eyestalk displacement of the interspecific hybrid crab. 

The most significant DAS was annotated as endoplasmic reticulum chaperone binding immunoglobin protein (BiP), also known as glucose regulatory protein 78 (GRP78) or HSPA5, and it is the major family member of heat shock protein 70 (Hsp70) that is required for protein folding and quality control in the endoplasmic reticulum [60]. BiP may stimulate the folding of freshly synthesized polypeptides and repair misfolded proteins to avoid their aggregation in the endoplasmic reticulum [61]. These findings suggested that the down-regulated expression of endoplasmic reticulum chaperone BiP may result in the aggregation of misfolded proteins and hence may lead to eyestalk displacement. In addition, previous studies showed that BiP plays an important role in the response to environmental stress [62,63] and immune function [64,65], suggesting that the down-regulated expression of endoplasmic reticulum chaperone BiP may impact the adaptability and disease resistance in DH. In the enrichment analysis, we found that most of the top ten significantly gene ontology terms were enriched with genes which are linked to cuticle-related or chitin-related functions. Similar results were also found in our previous study using whole-transcriptome RNA sequencing [15]. Cuticle protein is considered as the major component of the exoskeleton in crustaceans. Similarly, chitin is a key component of the cuticle that protects from external threats. A previous study has shown that cuticle-related genes play an essential role in normal wing morphogenesis in the migratory locust [66]. Therefore, the expression of cuticle-related or chitin-related genes such as *chitinase 7*, *chitin synthase*, *cuticle protein 21*, *cuticle protein 7-like*, and *early cuticle protein 2* (Appendix A), may play an essential role in eyestalk displacement. In addition, the downregulation of cuticle-related or chitin-related genes in DH would affect molting and consequently, growth and mating. One possible reason is that cuticular chitin synthase and chitinase are involved in the degradation of old cuticle and the synthesized of new cuticle during molting [67,68]. Moreover, in the KEGG enrichment analysis, the phototransduction pathway was also significantly enriched (*p* = 0.006) suggesting that eyestalk displacement would affect phototransduction function in DH, except for the top significantly enriched pathways involved in transcription, protein synthesis, protein transportation, and protein degradation (Figure 4D and Appendix A). Furthermore, the eyestalk is an essential phototransduction organ to receive light signals in crustaceans [69]. Overall, our findings provide valuable insights into the genetic mechanisms of eyestalk displacement and its potential implications on the novel interspecific hybrid crab, which would help enhance the genetic makeup of interspecific hybridization in the mud crab (*S. paramamosain*).

### 4.3. The Transcriptomes Difference between Purebred (SPA) and Crossbred (NH and DH) Mud Crabs

The completeness and quality of the reconstructed transcriptome is an important prerequisite aspect to reveal the genetic changes in novel interspecific hybrid crabs. In this study, the complete transcripts in BUSCO ranged from 38.7% to 48.4% for SPA, NH, and DH (Figure 1A), whereas the reference transcriptome was ~72.3% [28]. Moreover, the proportion of complete transcripts in BUSCO was 82.2% higher than in previous SMRT sequencing-based transcriptome in mud crabs (*S. paramamosain*) [49]. The high percentage of missing transcripts in this study might be attributed to the tissue specificity and spatiotemporal specificity [51,52], because all the samples originate from the same life stages (the first stage of the crablet). In addition, the proportion of encoded fragmented proteins in SPA, NH, and DH ranged from 1.4% to 1.9% (Figure 1A), which was lower than the reference transcriptome (3.0%) [28] and the previous SMRT sequencing-based transcriptome (3.2%) in mud crabs (*S. paramamosain*) [49], indicating the high quality of SMRT sequencing and data processing in this study. The characteristics analysis showed a high number of isoforms in these three reconstructed transcriptomes, especially in hybrid crabs (Figure 1C). The proportion of transcripts with more than two isoforms in purebred mud crabs (57.7% in SPA) was lower than in the hybrid crabs (67.9% in DH and 75.5% in NH), highlighting a high degree of transcriptome complexity (Figure 1C). Another study reported that the high degree of transcriptome complexity was the genetic basis of yield heterosis [70]. However, more abundant isoforms in hybrid crabs did not result in more alternative splicing events (Figure 3B), which contradicted a recent study that more alternative splicing events were detected in hybrids under heat stress and may contribute to heterosis in abalone [71]. These findings suggested that, in our study, hybridization might increase the isoforms of certain genes, while decreasing the isoforms of others. For example, the number of genes with more than six isoforms was 351 in NH, much more than SPA (191) (Figure 1C). Furthermore, the comparison between reconstructed transcriptomes and the reference genome annotation indicated that reconstructed transcriptomes contain more potentially novel or unknown transcript isoforms (Figure 1D). These results suggested that genes with increasing or decreasing isoforms may play an important role in the heterosis of hybrid crabs.

## 5. Conclusions

The use of SMRT and Illumina RNA sequencing to reconstruct transcriptomes based on the two-step collapsing strategy is an efficient way to identify candidate genes in the novel interspecific hybrid crab (*S. serrata* ♀ × *S. paramamosain* ♂). With the aim to disentangle the mechanisms related to eyestalk displacement, a total of 37 significant DAS events (17 up-regulated and 20 down-regulated) and 1475 significant DETs (492 up-regulated and 983 down-regulated) in DH were identified by differential expressed analysis based on the reconstructed transcriptome. The most significant DAS and DETs were annotated as being endoplasmic reticulum chaperone BiP and leucine-rich repeat protein lrrA-like isoform X2. Furthermore, the majority of the top ten significant GO terms were associated with cuticle or chitin development, suggesting that the expression of cuticle-related or chitin-related genes plays an essential role in eyestalk displacement. Overall, our findings provide valuable insights into the genetic mechanisms of eyestalk displacement and its potential impacts on the novel interspecific hybrid crab, which would help to improve the genetic improvement of interspecific hybridization in mud crabs (*Scylla* spp.).

## Figures and Tables

**Figure 1 biology-11-01026-f001:**
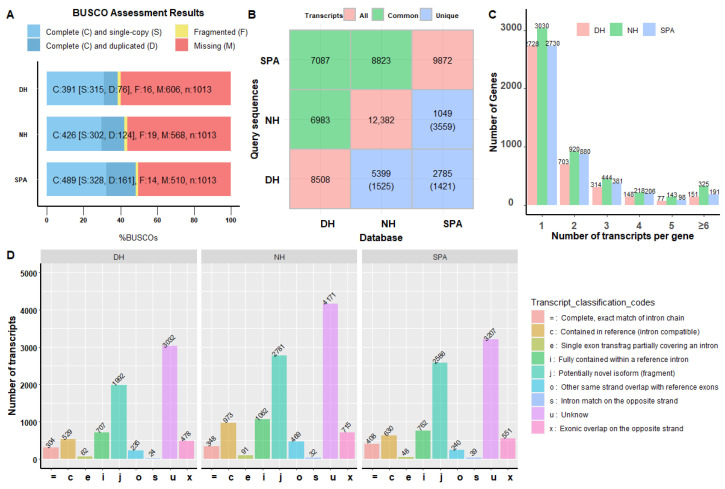
Completeness and characteristics analysis of reconstruction transcriptomes. (**A**) BUSCO assessment results of collapsed redundant transcripts. The Y-axis represents reconstructed transcriptome of different samples. Both the X-axis and the different colors of the box represents the proportion of different categories, including complete and single-copy, complete and duplicated, fragmented, or missing. (**B**) Common and unique transcripts among different transcriptomes. These diagonal, lower triangle, and upper triangle values are the number of transcript isoforms in the database, the number of unique transcript isoforms in the database (query sequences), and the number of common transcript isoforms between query sequences and the database. (**C**) The distribution of gene locus with different transcript isoforms number in SPA, NH, and DH. The Y-axis represents the number of genes. The X-axis represents gene locus with different isoforms. (**D**) Compared reconstructed transcriptomes with reference genome annotation using gffcompare v0.12.2 software. The Y-axis represents the number of genes. The X-axis and the different colors represent different categories.

**Figure 2 biology-11-01026-f002:**
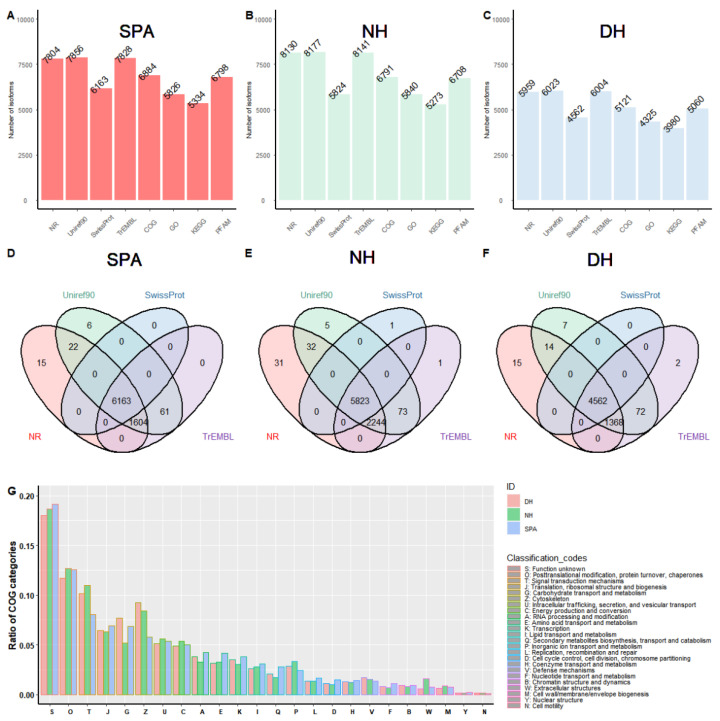
The summary of gene functional annotation using different databases. (**A**–**C**) Statistics of isoforms annotation results for SPA, NH, and DH using different databases including NR, Uniref90, Swiss-Prot, TrEMBL, COG, GO, KEGG, and PFAMs. The Y-axis represents the number of annotated isoforms. The X-axis represents different databases. (**D**–**F**) Venn diagrams showing the overlapping isoforms annotation results obtained using a different database for SPA, NH, and DH, respectively. (**G**) COG profiles of transcripts isoforms in SPA, NH, and DH.

**Figure 3 biology-11-01026-f003:**
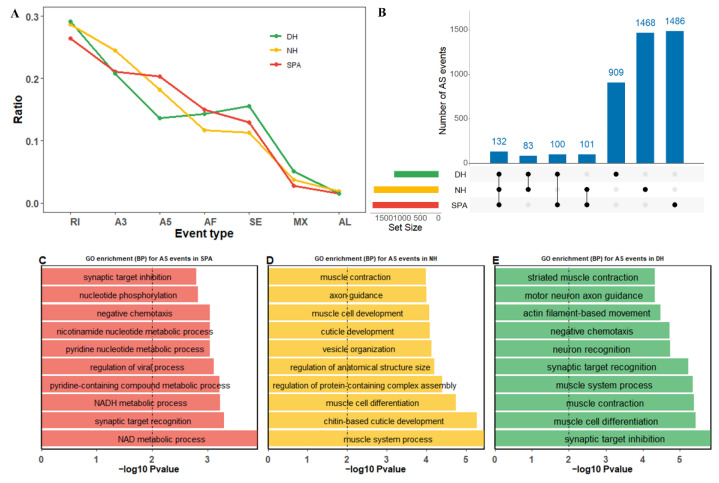
Summary of alternative splicing (AS) events profiling in SPA, NH, and DH. SPA, NH, and DH are indicated by different colors (red, yellow, and green, respectively). (**A**) The proportion of each AS type in SPA, NH, and DH. The Y-axis represents the proportion of different AS events. The X-axis represents different AS event types, including SE (skipped exon), MX (mutually exclusive exon), A5 (alternative 5′ splice site), A3 (alternative 3′ splice site), RI (retained intron), AF (alternative first exon), and AL (alternative last exon). (**B**) Identified common and specific AS events among SPA, NH, and DH. The barplots on the left represent the size of the datasets of SPA, NH, and DH. Dots and vertical lines indicate the overlapping AS events in the respective comparison. Barplots in the top panels represent the number of AS events. (**C**–**E**) The top ten significantly biological processes (BP) obtained from Gene Ontology (GO) enrichment analysis using genes with AS events in SPA, NH, and DH, respectively. The Y-axis represents different BP categories. The X-axis represents the corresponding −log10 transformed *p*-value.

**Figure 4 biology-11-01026-f004:**
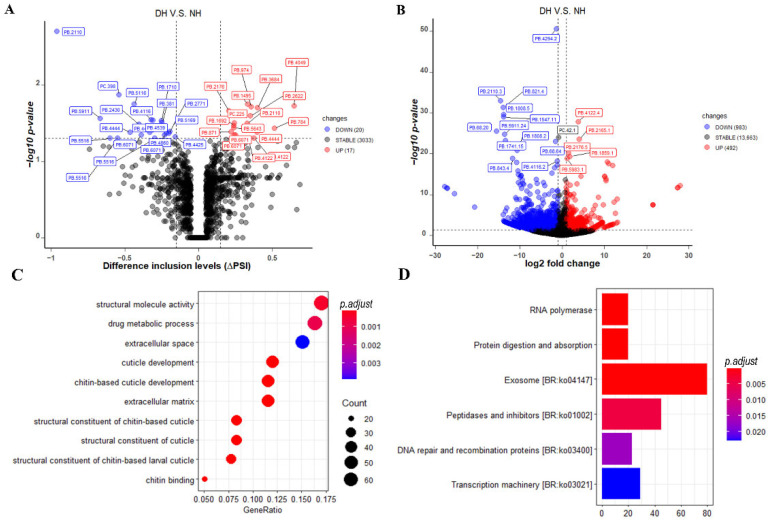
Differential expression analysis and enrichment analysis between DH and NH. (**A**) The volcano plot indicates *p*-values with minus log10-transformed for AS events (Y-axis) against their corresponding difference in inclusion levels (∆PSI) of each AS event (X-axis). The horizontal gray dotted line represents the significant threshold (0.05). The red, blue, and gray points represent up-regulated, down-regulated, and non-regulated AS events in DH groups, respectively. (**B**) The volcano plot indicates with minus log10-transformed for genes (Y-axis) against their corresponding log2(|fold change|) of echo gene (X-axis). (**C**) The top ten significant gene ontology (GO) terms obtained from GO enrichment analysis using genes with DAS events or DETs in SPA, NH, and DH, respectively. The Y-axis represents different GO term categories. The X-axis represents the proportion of significant expressed genes in the list of corresponding GO terms (GeneRatio). Different sizes and colors of circle represent the number of significantly expressed genes and corresponding adjusted *p*-value of GO terms. (**D**) The top ten significant pathways obtained from KEGG enrichment analysis using genes with DAS events or DEGs in SPA, NH, and DH, respectively. The Y axis represents different pathways categories. The X-axis represents the number of significantly expressed genes in the corresponding pathway. Different colors represent the different adjusted *p*-value of pathway.

**Figure 5 biology-11-01026-f005:**
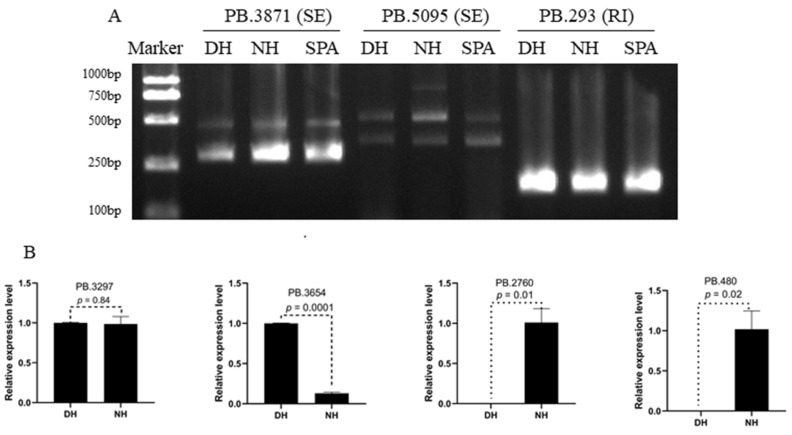
Validation of alternative splicing (AS) events and significant differential expression genes. (**A**) Validation of AS events by RT-PCR and agarose gel electrophoresis. (**B**) The relative expression level of PB.3297, PB.3654, PB.2760, and PB.480 in NH and DH.

**Table 1 biology-11-01026-t001:** Summary of PacBio sequencing data in SPA, NH, and DH.

Sample	Types	Numbers of Sequences	Length of Isoforms			N50 ^9^
			Min	Mean	Max	
SPA ^1^	Subreads	12,705,473	51	1636.8	211,052	1962
	CCS ^4^	265,745	104	1904	14,128	2177
	FL ^5^	226,748	99	1814.2	10,492	2109
	FLCN ^6^	221,222	68	1754.5	9608	2051
	HQ ^7^	11,643	72	1780.5	6442	2053
	LQ ^8^	14	321	1883.8	3922	1839
NH ^2^	Subreads	15,425,110	50	1491.4	115,220	1656
	CCS	420,743	67	1755.3	11,556	1939
	FL	345,977	102	1613.7	9605	1735
	FLCN	308,364	51	1495.2	9573	1534
	HQ	16,587	70	1630.5	6778	2012
	LQ	12	452	857.1	1617	951
DH ^3^	Subreads	15,459,279	50	1437.4	118,301	1590
	CCS	367,638	70	1782.2	14,497	1994
	FL	253,545	104	1683.4	11,344	1926
	FLCN	221,371	50	1568.1	9804	1769
	HQ	10,336	69	1661.6	6526	2132
	LQ	9	115	604.1	1880	654

^1^ SP = *Scylla paramamosain* with normal eyestalk; ^2^ NH = hybrid crabs with normal eyestalk; ^3^ DH = hybrid crabs with displaced eyestalk; ^4^ CCS = circular consensus sequence; ^5^ FL = full-length; ^6^ FLNC = full-length-non-chimeric; ^7^ HQ = high-quality isoforms; ^8^ LQ = Low-quality isoforms; ^9^ N50 = 50% of reads are longer than this value.

**Table 2 biology-11-01026-t002:** Summary of features of transcript isoforms after collapsing redundant isoforms with cDNA cupcake, cogent, and CD-HIT.

Samples	Numbers of Transcript Isoforms after Collapsing Redundant Isoforms				Length of Collapsing Redundant Isoforms			N50 ^4^
	Reference Genome	Fake Genome	Unmap-Ped	Merge	Min	Max	Mean	
SPA ^1^	9427	466	2	9872	92	6442	1798.9	2079
NH ^2^	11,639	752	3	12,382	118	6778	1663.6	2041
DH ^3^	7858	661	3	8508	80	6526	1685.5	2161

^1^ SPA = *Scylla paramamosain* with normal eyestalk; ^2^ NH = hybrid crabs with normal eyestalk; ^3^ DH = hybrid crabs with deformed eyestalk; ^4^ N50 = 50% of reads are longer than this value.

## Data Availability

The datasets presented in this study can be found in online repositories. The SMRT sequencing data of SPA, NH, and DH could be found in NCBI with the accession number SRX9982023, SRX10000714, and SRX10000713, respectively. The RNA-seq data also stored in NCBI with the accession number (PRJNA805205). In addition, the reconstructed transcriptomes of SPA, NH, DH, and the novel interspecific hybrid crab have been uploaded to the figshare repository (https://doi.org/10.6084/m9.figshare.19144988.v1, accessed on 9 February 2022).

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
