# Peer review of "Full-Length Transcriptome Reconstruction Reveals the Genetic Mechanisms of Eyestalk Displacement and Its Potential Implications on the Interspecific Hybrid Crab (Scylla serrata ♀ × S. paramamosain ♂)"

_biology, 2022, doi:10.3390/biology11071026_

Round 1
Reviewer 1 Report
The manuscript presents an interesting study and interesting results about genetic mechanisms of eyestalk displacement in a novel interspecific hybrid crab.
The experimental procedures and analyses were conducted properly. I have some comments and suggestions mainly on the formal aspect of the manuscript.
“Material and methods” section is very dense; if you can, smooth it to facilitate the reader.
In “Results” section, pay attention to presents the results objectively, be careful to not include sentences aimed at the interpretation of the results, that instead should go in the “Discussion” section (see the point-by-point revision).
In “Discussion”, is not necessary to repeat the results and the indication of Figures and Tables is not appropriate for this section (see the point-by-point revision).
Try to be consistent throughout the paper when you indicate the p-value; see for example line 219, 222, 228, 468-470, 533, every time p-value is indicated in a different way.
Pay attention to the scientific names of the species, they have to be indicated in italics every time (see lines 19, 29, 71).
Figure captions: they have to be self-explicative, but it is not necessary to give material and methods details here, just explain what is represented in the figure.
Please, see the following point-by-point revision.
SIMPLE SUMMARY
Line17: Remove however
Line 20: implications
Line 21: remove “for further analysis”
Line 22: I would start the sentence from “37 significantly differential..”
ABSTRACT
Line 34-35: this sentence is not clear, please rephrase
Line 40-41: as in line 22
INTRODUCTION
Line 63: most of the studies
Line 64: nutritional composition à what do you mean? Chemical composition of meat?
Line 66: a common and effective method
Line 70-72: please rephrase, it is not clear
Line 82: remove in summary
Line 98: and others à remove “others” or add references referred to others
Line 103: crabs
Line 104: Remove respectively
MATERIALS AND METHODS
Line 114-115: move “in our previous study” after was developed
Please describe briefly the procedure
Line 122: remove “the”
Qualified RNAs à you mean “high quality”?
Line 123: remove “in this study”
Line 126: remove “respectively”
Line 126-127: It is not clear, please rephrase
Line 129-130: you mean three replicates consisting of one abnormal hybrid crab + two normal hybrid crabs each replicate? Please specify
Line 133: were processed
Line 134, 137, 138, 141, 142: paraments? I think it is parameters
Line 155: was created
Line 161: BUSCO, this is the first appearance, please use the extended (benchmarking universal single-copy orthologs)
Line 162: using BUSCO
Line 167-168: coded as (not code as)
Line 168: remove “and others”
Line 169: Put a full a stop and start with “The detailed information can be found on the official website…”
Line 172: applying the following parameters
Line 173-175: not completely clear
Line 179: use the extended for ORF at first appearance
Line 202: following the above-mentioned steps
RESULTS
Line 256: was minimal that we could..
Line 258: please rephrase
Line 259-262: use superscript for the numbers
Line 260: displaced eyestalk
Table 2: better to leave a space between “number of transcript isoform” and “length of collapsing”; it is not clear that “Merge” belongs to the first group
Line 277: maybe “Summary of features of transcript isoforms…”
Line 279-280: use superscript for the numbers
Line 280: displaced eyestalk
Line 292: it is 7087 not 7067
6983 and 7087
When DH were compared
When SPA were compared with NH
Line 293-295: The number of unique transcript isoforms was 1525 or 1421 293 when DH were compared to NH or SPA, 5399 or 3559 when NH were compared to DH or SPA, and 2785 or 1421 when SPA were compared to DH or NH.
Line 295: is it 1421? From the figure I see 1049
Line 295-297: written like this, it seems more a “Discussion” sentence
Line 301-304: please rephrase dividing novel isoforms and unknown isoforms
Line 303: coded
Line 305-306: not clear
Line 306: add the number before percentage for DH
Line 307-309: this is a “Discussion” sentence
Line 313: reconstructed transcriptome maybe?
Line 316: remove “identified”
Line 317-318: it is not necessary to give material and methods details here, just explain what is represented in the figure
Line 345-348: “Discussion” sentence
Line 351: results for SPA, NH and DH
Line 351-352: when you list the databases, follow the same order as the figure
Line 352: remove “respectively”
Line 353: diagrams
Line 354: showing the overlapping isoforms annotation results obtained using different databases for SPA, NH, and DH, respectively.
Line 365: Remove from “in comparison” to “DH” and start directly from “A total”
Line 380-384: “Discussion” sentence
Line 386: remove “the” at the beginning
Line 387: SPA, NH and DH are indicated by different colors (red, yellow and green, respectively)
Line 397: -log10 transformed p-value
Line 436: adjusted p-value
Line 437-439: “Discussion” sentence
Line 442 and 446: -log10 transformed p-value
Line 452,455: adjusted p-value
Line 455: colors
Line 455-456: not clear, please rephrase
Line 472-473: “Discussion” sentence
DISCUSSION
Line 479-480: not sure that interspecific hybridization is one of the most common and effective breeding technologies in aquatic animals. Selective breeding, genetic and genomic selection with the aim of enhancing several traits, are valuable and effective breeding technologies as well, and they are very common in aquatic animals, probably more than interspecific hybridization.
What should be underlined here are the advantages of hybridization for the study species and why this technology is better compared to other technologies.
Line 482: a novel hybrid mud crab
Line 483: genetic composition is very vague, what do you mean? Also, please specify and underline the importance of this novel species (in which aspects is better?)
Line 483-484: not clear, please rephrase
Line 499 and after: indication of figures and tables are not necessary here, this is a discussion part
Line 503: as in the introduction, and others à remove “others” or add references referred to others
Line 504: In other species, such as Drosophila, … (remove "however", remove "of Drosphila" at the end)
Line 506: Remove “as a result”; “We can conclude”
Line 508: use the extended for BiP
Line 517: environmental stress
Line 520: better gene ontology
Line 521: two also in the same sentence
Line 526-528: move “such as chitinase 7, chitin synthase, cuticle protein 21, 527 cuticle protein 7-like, early cuticle protein 2” after “cuticle-related or chitin-related genes”; remove “so on”
Line 530-532: Please rephrase
Line 536-537: the sentence seems incomplete, please check
Line 549: spatiotemporal specificity, what do you mean? Is it related to the life stage?
Line 550-551: is this sentence related to the previous one?
Paragraph 4.3: maybe this paragraph should go as first
CONCLUSION
Maybe declare that the aim was to disentangle the mechanisms related to eyestalk displacement
Line 616: again, genetic composition is very vague, be more specific
Author Response
Dear reviewer 1,
Thank you very much for comments and suggestions. According to your comments and suggestions, we revised our manuscript completely and carefully. Now, we replied the comments point by point as follows:
Line17: Remove however
Author: Thanks, modified.
Line 20: implications
Author: Thanks, modified.
Line 21: remove “for further analysis”
Author: Thanks, modified.
Line 22: I would start the sentence from “37 significantly differential..”
Author: Thanks, modified.
ABSTRACT
Line 34-35: this sentence is not clear, please rephrase
Thanks, this sentence has been rewritten. The new sentence was: “Meanwhile, an interesting phenomenon was discovered that some first-generation (F1) hybrid offspring's eyestalks were displaced during the crablet stage I”, please see Line 33-34.
Line 40-41: as in line 22
Author: Thanks, modified.
INTRODUCTION
Line 63: most of the studies
Author: Thanks, modified.
Line 64: nutritional composition à what do you mean? Chemical composition of meat?
Author: Yes, just like the nutritional composition of meat. For expamle, the type, number and proportion of amino acids they contain.
Line 66: a common and effective method
Author: Thanks, modified.
Line 70-72: please rephrase, it is not clear
Author: Thanks, this sentence has been rewritten. The new sentence was: “However, interspecific hybridization in mud crab (Scylla spp.) not only bring hybrid vigor but also hybrid inferiority, because some hybrid offspring had eyestalks displacement during the crablet stage I”. please see Line 68-70.
Line 82: remove in summary
Author: Thanks, modified.
Line 98: and others à remove “others” or add references referred to others
Author: Thanks, modified.
Line 103: crabs
Author: Thanks, modified.
Line 104: Remove respectively
Author: Thanks, modified.
MATERIALS AND METHODS
Line 114-115: move “in our previous study” after was developed
Please describe briefly the procedure
Author: Thank you for your suggestion. We describe this procedure briefly as follow: The male S. paramamosain was obtained from the local shore (Guangdong province, China), and the female S. serrata was bought from a local market in Shantou, China. The artificial mating of Scylla was performed following the method described by our previous study [13]. After artificial mating, female crabs were transferred to a rearing tank until the egg hatch. The F1 hybrid larvae were hatched in circular fiberglass tanks (0.9 m diameter, 1.0 m height), and then transferred to concrete rearing tanks (5.8 m × 4.8 m × 1.8 m) for grow seedlings. Culture conditions were ambient temperature (almost 30 â—¦C), natural photoperiod, and salinity of approximately 30 ppt. Please see Line 111-119
Line 122: remove “the”
Qualified RNAs à you mean “high quality”?
Author: Yes, modified.
Line 123: remove “in this study”
Author: Thanks, modified.
Line 126: remove “respectively”
Author: Thanks, modified.
Line 126-127: It is not clear, please rephrase
Author: Thanks, this sentence has been rewritten. The new sentence was: “In addition, each SMRT library included pooled RNAs from three crab samples”, please see Line 130.
Line 129-130: you mean three replicates consisting of one abnormal hybrid crab + two normal hybrid crabs each replicate? Please specify
Author: No, we constructed three libraries per group for DH and NH group, but libraries contained different number of samples in different group. The sentence was rewrite as: “For Illumina sequencing, three libraries of DH group (one abnormal hybrid crab per library) and three libraries of NH group (two normal hybrid crabs per library) were constructed and sequenced on the Illumi-na Hiseq 2500 platform to generate 150bp paired-end reads”, please see Line131-133.
Line 133: were processed
Author: Thanks, modified.
Line 134, 137, 138, 141, 142: paraments? I think it is parameters
Author: Thanks, modified.
Line 155: was created
Author: Thanks, modified.
Line 161: BUSCO, this is the first appearance, please use the extended (benchmarking universal single-copy orthologs)
Author: Thanks, modified.
Line 162: using BUSCO
Author: Thanks, modified.
Line 167-168: coded as (not code as)
Author: Thanks, modified.
Line 168: remove “and others”
Author: Thanks, modified.
Line 169: Put a full a stop and start with “The detailed information can be found on the official website…”
Author: Thanks, modified.
Line 172: applying the following parameters
Author: Thanks, modified.
Line 173-175: not completely clear
Author: Thanks, this sentence has been rewritten. The new sentence was: “the common and unique transcripts between transcriptomes were identified by turning one transcriptome as blast database and the other transcriptome as query sequence” please see Line177-180.
Line 179: use the extended for ORF at first appearance
Author: Thanks, modified.
Line 202: following the above-mentioned steps
Author: Thanks, modified.
RESULTS
Line 256: was minimal that we could..
Author: Thanks, modified.
Line 258: please rephrase
Author: Thank you for your suggestion, this sentence has been rewritten. The new sentence was: “Summary of PacBio sequencing data in SPA, NH, and DH”, please see Line 263.
Line 259-262: use superscript for the numbers
Author: Thanks, modified.
Line 260: displaced eyestalk
Author: Thanks, modified.
Table 2: better to leave a space between “number of transcript isoform” and “length of collapsing”; it is not clear that “Merge” belongs to the first group
Author: Thanks, modified.
Line 277: maybe “Summary of features of transcript isoforms…”
Author: Thanks, modified.
Line 279-280: use superscript for the numbers
Author: Thanks, modified.
Line 280: displaced eyestalk
Author: Thanks, modified.
Line 292: it is 7087 not 7067
6983 and 7087
When DH were compared
When SPA were compared with NH
Author: Thanks, modified.
Line 293-295: The number of unique transcript isoforms was 1525 or 1421 when DH were compared to NH or SPA, 5399 or 3559 when NH were compared to DH or SPA, and 2785 or 1421 when SPA were compared to DH or NH.
Author: Thanks, modified. Please see Line 297-300.
Line 295: is it 1421? From the figure I see 1049
Author: Thanks, modified.
Line 295-297: written like this, it seems more a “Discussion” sentence
Author: Thank you for your suggestion. This sentence has been rewritten as “In a word, the number and diversity of transcript isoforms in NH samples were greater than in other samples, and that most transcript isoforms of DH (82%) and SPA (89%) were similar with NH. Moreover, these reconstructed transcriptomes had numerous isoforms, with the number of transcripts with more than two isoforms were 1756, 2050 and 1393 in SPA, NH, and DH, respectively (Fig.1C). Please see Line 300-304.
Line 301-304: please rephrase dividing novel isoforms and unknown isoforms
Author: Thank you for your suggestion. This sentence has been rewritten as “In comparison to the reference genome annotation, a total of 1992, 2781 and 2566 potentially novel isoforms (coded as j) and a total of 3032, 4171 and 3297 unknown isoforms (coded as u) were annotated in reconstructed transcriptomes of DH, NH, and SPA, showed that the reconstructed transcriptomes contain more novel isoforms or transcripts”, please see Line 304-308.
Line 303: coded
Author: Thanks, modified.
Line 305-306: not clear
Author: Thank you for your suggestion. This sentence has been rewritten as “In addition, the number of transcripts isoforms that were annotated as other same strand overlaps with reference exons (coded as o) was 469 (3.8%) in NH, which was higher than DH 235 (2.8%) and SPA 240 (2.5%), please see Line 308-310.
Line 306: add the number before percentage for DH
Author: Thanks, modified.
Line 307-309: this is a “Discussion” sentence
Author: Thanks, removed.
Line 313: reconstructed transcriptome maybe?
Author: Thanks, modified.
Line 316: remove “identified”
Author: Thanks, modified.
Line 317-318: it is not necessary to give material and methods details here, just explain what is represented in the figure
Author: Thanks, removed.
Line 345-348: “Discussion” sentence
Author: Thanks, removed.
Line 351: results for SPA, NH and DH
Author: Thanks, modified.
Line 351-352: when you list the databases, follow the same order as the figure
Author: Thanks, removed.
Line 352: remove “respectively”
Author: Thanks, removed.
Line 353: diagrams
Author: Thanks, modified.
Line 354: showing the overlapping isoforms annotation results obtained using different databases for SPA, NH, and DH, respectively.
Author: Thanks, modified.
Line 365: Remove from “in comparison” to “DH” and start directly from “A total”
Author: Thanks, removed.
Line 380-384: “Discussion” sentence
Author: Thanks, modified.
Line 386: remove “the” at the beginning
Author: Thanks, modified.
Line 387: SPA, NH and DH are indicated by different colors (red, yellow and green, respectively)
Author: Thanks, modified.
Line 397: -log10 transformed p-value
Author: Thanks, modified.
Line 436: adjusted p-value
Author: Thanks, modified.
Line 437-439: “Discussion” sentence
Author: Thanks, modified.
Line 442 and 446: -log10 transformed p-value
Author: Thanks, modified.
Line 452,455: adjusted p-value
Author: Thanks, modified.
Line 455: colors
Author: Thanks, modified.
Line 455-456: not clear, please rephrase
Author: Thank you for your suggestion. This sentence has been rewritten as “Different colors represent different the adjusted p-value of pathway”, please see Line 447-448.
Line 472-473: “Discussion” sentence
Author: Thanks, removed.
DISCUSSION
Line 479-480: not sure that interspecific hybridization is one of the most common and effective breeding technologies in aquatic animals. Selective breeding, genetic and genomic selection with the aim of enhancing several traits, are valuable and effective breeding technologies as well, and they are very common in aquatic animals, probably more than interspecific hybridization.
What should be underlined here are the advantages of hybridization for the study species and why this technology is better compared to other technologies.
Author: Thank you for your suggestion. This sentence has been rewritten as “Interspecific hybridization, which is well known for developing hybrid offspring with greater biomass, speed of development, and fertility, is considered one of the ef-fective breeding technologies in aquatic animals [11]”, please see Line 470-472
Line 482: a novel hybrid mud crab
Author: Thanks, modified.
Line 483: genetic composition is very vague, what do you mean? Also, please specify and underline the importance of this novel species (in which aspects is better?)
Author: Thank you for your suggestion. This sentence has been rewritten as “In our earlier work, we successfully developed a novel hybrid mud crab (S. serrata♀×S. paramamosain♂) for obtaining high-quality hybrid offspring for genetic improvement [13,14]”, please see Line 472-474.
Line 483-484: not clear, please rephrase.
Author: Thank you for your suggestion. This sentence has been rewritten as “Meanwhile, we found that some F1 hybrid offspring's eyestalks had displaced during the crablet stage I”, please see Line 474-475.
Line 499 and after: indication of figures and tables are not necessary here, this is a discussion part
Author: Thanks, removed.
Line 503: as in the introduction, and others à remove “others” or add references referred to others
Author: Thanks, modified.
Line 504: In other species, such as Drosophila, … (remove "however", remove "of Drosphila" at the end)
Author: Thanks, modified.
Line 506: Remove “as a result”; “We can conclude”
Author: Thanks, modified.
Line 508: use the extended for BiP
Author: Thanks, modified.
Line 517: environmental stress
Author: Thanks, modified.
Line 520: better gene ontology
Author: Thanks, modified.
Line 521: two also in the same sentence
Author: Thanks, removed.
Line 526-528: move “such as chitinase 7, chitin synthase, cuticle protein 21, 527 cuticle protein 7-like, early cuticle protein 2” after “cuticle-related or chitin-related genes”; remove “so on”
Author: Thanks, modified.
Line 530-532: Please rephrase
Author: Thank you for your suggestion. This sentence has been rewritten as “One possible reason is that cuticular chitin synthase and chitinase are involved in the degradation of old cuticle and the synthesized of new cuticle during molting”, please see Line 552-554.
Line 536-537: the sentence seems incomplete, please check
Author: Thank you for your suggestion. This sentence has been rewritten as “Because the eyestalk is an essential phototransduction organ to receive light signals in crustaceans”, please see Line 558-559.
Line 549: spatiotemporal specificity, what do you mean? Is it related to the life stage?
Author: Yes. This sentence has been rewritten as “The high percentage of missing transcripts in this study might be attributed to the tissue specificity and spatiotemporal specificity [57,58], because all the samples origi-nate from the same life stages (the first stage of crablet)”, please see Line 570-573.
Line 550-551: is this sentence related to the previous one?
Author: Yes. This sentence has been rewritten as “The high percentage of missing transcripts in this study might be attributed to the tissue specificity and spatiotemporal specificity [57,58], because all the samples origi-nate from the same life stages (the first stage of crablet)”, please see Line 570-573.
Paragraph 4.3: maybe this paragraph should go as first
Author: Thanks, modified.
CONCLUSION
Maybe declare that the aim was to disentangle the mechanisms related to eyestalk displacement
Author: Thanks, modified.
Line 616: again, genetic composition is very vague, be more specific
Author: Thanks, modified.

Reviewer 2 Report
In the current manuscript, the authors studied eyestalk displacement in mud crab hybrids at the transcriptome level. They compared the transcriptomes of hybrid crabs with displaced eyestalk, hybrid crabs with normal eyestalk, and purebred mud crab, and differentially expressed genes were identified. In general, the study is valuable and provides useful knowledge on the molecular mechanisms of eyestalk displacement. However, I have some comments for the authors.
The three groups used in this study were normal hybrids (NH), displaced hybrids (DH), normal purebred mud crab (S. paramamosain) (SPA). Why did not the authors sample the female parent (S. serrata)? The female parent will provide maternal inheritance and it may worth studying in this species. It may be too late to address this point but at least a justification would be good.
To validate the AS events, why did the authors verify the results using agarose gel electrophoresis only? Why DNA sequencing of amplicons (using Sanger method) was not used?
Other minor comments and suggestions:
Line 19: Simple summary: Scientific names should be italicized. Also, line 29.
Line 22: “expressed” should be “expression”. differential expression analysis. Also, line 41.
Line 29-30: “cross-breed” should be “crossbreeding”. Also, lines 49 and 111.
Line 35: eyestalks were displaced
Line 62: Check the structure of this sentence.
Line 63: Most of studies …. Delete “of”
Line 66-67: Add an article, “in” should be “of”, delete the comma, “Interspecific hybridization is a common ………… of aquatic animals [11] because …... “
Line 76: Delete the comma. “…… physiological processes involving ……”
Line 80: “increasing” should be “increased”.
Line 82: plays
Line 96: crustaceans
Line 97: “… patterns of specific tissues ……”
Line 100: delete “to”. “help uncover”
Line 104: There is no need for “respectively” here. Please delete it.
Line 114: “The F1 hybrid offspring were developed ………”
Line 120: Furthermore, the RNA quality was ………
Line 126-127: “…. each SMRT library included pooled RNAs from three crab samples.”
Line 155: “would be created” should be “was created”
Lines 154-157: Please provide a reference for using the “fake genome”.
Line 160: 2.4. Completeness and characteristics analysis of reconstructed transcriptomes
Line 204: reconstructed transcriptomes. Also, line 208.
Line 233: Primer pairs …
Line 256: ……… isoforms was ……
Line 285: The definition of BUSCO should be moved to the first time the abbreviation is used (line 161)
Line 293: … transcript isoforms for DH ………… Also, lines 294 and 295, “when” should be “for”.
Line 327: To obtain the biological context of the reconstructed ……..
Line 329: The definition of ORF should be moved to the first place the abbreviation is used (line 179).
Line 441: Differential expression analysis ……..
Line 509: Add a comma after “BiP)”.
Line 516: plays
Line 521: similar results were also found ………
Line 530: This sentence “This may because of that during molting ….. “. is not clear. Please rewrite.
Line 536-537: This sentence is grammatically incomplete. Please check.
Line 540: help enhance
Line 550: The sentence beginning with “Because …” is incomplete. Please check.
Line 638: Please italicize scientific names.
References: Please be consistent when citing journal names. For example, some journal names are in the “Sentence case” such as reference # 4, while for others, “Each Word Is Capitalized” such as Reference # 2.
Also, the scientific names should be italicized in the title of articles in the references list too.
Author Response
Dear reviewer2,
Thank you very much for comments and suggestions. According to your comments and suggestions, we revised our manuscript completely and carefully. Now, we replied the comments point by point as follows:
Major points:
The three groups used in this study were normal hybrids (NH), displaced hybrids (DH), normal purebred mud crab (S. paramamosain) (SPA). Why did not the authors sample the female parent (S. serrata)? The female parent will provide maternal inheritance and it may worth studying in this species. It may be too late to address this point but at least a justification would be good.
Author: Thanks. we agree with your viewpoint. When we design this experiment, we plan to cultivate seedlings of purebred mub crabs (S. serrata), purebred mub crabs (S. paramamosain), and the novel hybrids mud crab. However, only purebred mub crabs (S. paramamosain) and hybrid larvae were hatched and grow to the first stage of crablet. Because the larvae of purebred mub crabs (S. serrata) are not adapted the local climate and environment.
To validate the AS events, why did the authors verify the results using agarose gel electrophoresis only? Why DNA sequencing of amplicons (using Sanger method) was not used?
Author: Thank you for your suggestion. Just to validate the AS events whether truthly exist in samples, we think the result of agarose gel electrophoresis of RT-PCR products is enough to reflect directly. The similar practice also could be found in previous study (Tang et al. 2018).
Tang, L. T., X. Q. Ran, N. Mao, F. P. Zhang, X. Niu, Y. Q. Ruan, F. L. Yi, S. Li, and J. F. Wang. 2018. 'Analysis of alternative splicing events by RNA sequencing in the ovaries of Xiang pig at estrous and diestrous', Theriogenology, 119: 60-68.
Other minor comments and suggestions:
Line 19: Simple summary: Scientific names should be italicized. Also, line 29.
Author: Thanks, modified.
Line 22: “expressed” should be “expression”. differential expression analysis. Also, line 41.
Author: Thanks, modified.
Line 29-30: “cross-breed” should be “crossbreeding”. Also, lines 49 and 111.
Author: Thanks, modified.
Line 35: eyestalks were displaced
Author: Thanks, modified.
Line 62: Check the structure of this sentence.
Author: Thanks, this sentence has been rewritten as: “Furthermore, the genetic improvement of mud crab (S. paramamosain) is still in its in-fancy compared with other aquatic species”. Please see Line 59-61.
Line 63: Most of studies …. Delete “of”
Author: Thanks, modified.
Line 66-67: Add an article, “in” should be “of”, delete the comma, “Interspecific hybridization is a common ………… of aquatic animals [11] because …... “
Author: Thanks, modified.
Line 76: Delete the comma. “…… physiological processes involving ……”
Author: Thanks, modified.
Line 80: “increasing” should be “increased”.
Author: Thanks, modified.
Line 82: plays
Author: Thanks, modified.
Line 96: crustaceans
Author: Thanks, modified.
Line 97: “… patterns of specific tissues ……”
Author: Thanks, modified.
Line 100: delete “to”. “help uncover”
Author: Thanks, modified.
Line 104: There is no need for “respectively” here. Please delete it.
Author: Thanks, modified.
Line 114: “The F1 hybrid offspring were developed ………”
Author: Thanks, modified.
Line 120: Furthermore, the RNA quality was ………
Author: Thanks, modified.
Line 126-127: “…. each SMRT library included pooled RNAs from three crab samples.”
Author: Thanks, modified.
Line 155: “would be created” should be “was created”
Author: Thanks, modified.
Lines 154-157: Please provide a reference for using the “fake genome”.
Author: Thank you for yor suggestion. The "fake genome" was created by concatenating all cogent unassigned contigs from SMRT sequencing data, did not come from other studies.
Line 160: 2.4. Completeness and characteristics analysis of reconstructed transcriptomes
Author: Thanks, modified.
Line 204: reconstructed transcriptomes. Also, line 208.
Author: Thanks, modified.
Line 233: Primer pairs …
Author: Thanks, modified.
Line 256: ……… isoforms was ……
Author: Thanks, modified.
Line 285: The definition of BUSCO should be moved to the first time the abbreviation is used (line 161)
Author: Thanks, modified.
Line 293: … transcript isoforms for DH ………… Also, lines 294 and 295, “when” should be “for”.
Author: Thanks, modified.
Line 327: To obtain the biological context of the reconstructed ……..
Author: Thanks, modified.
Line 329: The definition of ORF should be moved to the first place the abbreviation is used (line 179).
Author: Thanks, modified.
Line 441: Differential expression analysis ……..
Author: Thanks, modified.
Line 509: Add a comma after “BiP)”.
Author: Thanks, modified.
Line 516: plays
Author: Thanks, modified.
Line 521: similar results were also found ………
Author: Thanks, modified.
Line 530: This sentence “This may because of that during molting ….. “. is not clear. Please rewrite.
Author: Thanks, this sentence has been rewritten as: “One possible reason is that cuticular chitin synthase and chitinase are involved in the degradation of old cuticle and the synthesized of new cuticle during molting”, please see Line 552-554.
Line 536-537: This sentence is grammatically incomplete. Please check.
Author: Thank you for your suggestion. This sentence has been rewritten as “Because the eyestalk is an essential phototransduction organ to receive light signals in crustaceans”, please see Line 558-559.
Line 540: help enhance
Author: Thanks, modified.
Line 550: The sentence beginning with “Because …” is incomplete. Please check.
Author: Thanks, modified.
Line 638: Please italicize scientific names.
Author: Thanks, modified.
References: Please be consistent when citing journal names. For example, some journal names are in the “Sentence case” such as reference # 4, while for others, “Each Word Is Capitalized” such as Reference # 2.
Author: Thanks, modified.
Also, the scientific names should be italicized in the title of articles in the references list too.
Author: Thanks, modified.
